# Neural Fingerprints for Adversarial Attack Detection

## Abstract

Deep learning models for image classification have become standard tools in recent years. However, a well known vulnerability of these models is their susceptibility to adversarial examples. Adversarial examples are generated by slightly altering an image of a certain class in a way that is imperceptible to humans but causes the model to classify it wrongly as another class. Many algorithms have been proposed to address this problem, falling generally into one of two categories: (i) building robust classifiers (ii) directly detecting attacked images. Despite the very good performance of the proposed detectors, we argue that in a white-box setting, where the attacker knows the configuration and weights of the network and the detector, the attacker can overcome the detector by running many examples on a local copy, and sending only examples that were not detected to the actual model. This problem of addressing complete knowledge of the attacker is common in security applications where even a very good model is not sufficient to ensure safety. In this paper we propose to overcome this inherent limitation of any static defence with randomization. To do so, one must generate a very large family of detectors with consistent performance, and select one or more of them randomly for each input. For the individual detectors, we suggest the method of neural fingerprints. In the training phase, for each class we repeatedly sample a tiny random subset of neurons from certain layers of the network, and if their average is sufficiently different between clean and attacked images of the focal class they are considered a fingerprint and added to the detector bank. During test time, we sample fingerprints from the bank associated with the label predicted by the model, and detect attacks using a likelihood ratio test. We evaluate our detectors on ImageNet with different attack methods and model architectures, and show near-perfect detection with low rates of false detection.

## 1 Introduction

In recent years, deep learning models have become ubiquitous across a wide range of applications, from image classification and natural language processing to speech recognition and transportation. However, these models have been shown to be vulnerable to adversarial attacks - small, imperceptible perturbations to inputs that cause models to make incorrect predictions (Szegedy et al., 2014; Goodfellow et al., 2014). These attacks pose serious concerns regarding the security and reliability of deep learning systems, especially in critical domains (Kurakin et al., 2016). A growing body of research has focused on developing adversarial attacks that can reliably fool models as well as defences to mitigate these threats (Madry et al., 2017; Song et al., 2017; Gowal et al., 2020).

The main defence approaches can be divided into the broad categories of robust classification and adversarial attack detection. As the name suggests, robust models aim to mitigate the threat by being

robust to such inputs. This is often achieved by introducing training schemes that include adversarial examples, or by alterations of the inputs aiming to negate the adversarial perturbation. Alternatively, in adversarial attack detection, the main model is used as-is, but a parallel model performs the binary classification of the input as clean or attacked. This is done for instance by adding and training an additional output head from one of the layers of the network, or via statistical models that consider network activations or the final output layer. A short introduction to the main adversarial attack and protection methods is given in Section (2).

When considering the truly white-box threat model – that is, assuming that the attacker has complete knowledge of the system – even near-perfect detection will not suffice. Knowing the structure and parameters used both for the main classifier and for the detector model, the attacker need only run many attack attempts on an offline copy of the system, and present the actual system only with inputs that were already verified to be successful adversarial attacks. If the detector model is differentiable, as is the case when adding an extra binary output head, the attacker can feasibly bypass the defence even more directly, by adding the desired (negative) response of the detector model to the objective when computing the perturbation for the adversarial attack.

Imagine however if we could have multiple detector networks, each providing a consistent and acceptable detection level. During inference, we randomly choose one of these detectors to apply to the input. This randomized strategy prevents users from crafting an input that could compromise both the network and the detector, as they won't know which detector will be selected. For this method to be effective, we need: (i) a large pool of detectors to choose from, and (ii) detectors that are not highly correlated, so that attacking one does not affect many others. Additionally, the entire process must be computationally efficient to ensure practicality.

In this paper, we introduce the concept of *Neural Fingerprints*. A neural fingerprint consists of a subset of neurons with a known distributions given a specific class. We demonstrate that grouping just a few dozen neurons into a fingerprint can achieve considerable detection rates, and by using many fingerprints together we can achieve near-perfect detection with a negligible false alarm rate. Additionally, we present an efficient method to prepare a large bank of fingerprints that share very few neurons, allowing for the selection of an uncorrelated random subset of fingerprints at test time. Our method is validated on the ImageNet dataset, where we systematically created adversarial attacks across classes. This extensive experimentation surpasses that of most studies in adversarial detection methods, suggesting the practical effectiveness and scalability of our method in real-world scenarios.

The intuition behind the use of neural fingerprints for adversarial attack detection relies on several facts to hypothesize that such detectors would exist in many cases. First, from the lottery ticket hypothesis (Frankle & Carbin, 2018), we know that most neurons are not actively driving the classification result. Moreover, the various adversarial attacks try to make as little change as possible, and hence will mostly change the value of the activations that do affect the classification. From this we conclude that most neurons in a random set of neurons will not be significantly impacted by an adversarial attack. Finally, due to the way that networks are trained (e.g., small gradient steps, dropout) many neurons that are not currently important for the classification do carry some information about the class that was gathered during training. In total, we hypothesise that information about the identity of the true class is distributed among a large population of neurons, most of which are not highly influential in the classification output and thus they will not be targeted by adversarial attacks. We attempt to extract and exploit this information in our detectors.

The main contribution of this paper is twofold. First, to the best of our knowledge this paper is the first to address the insufficiency of deterministic adversarial attack detectors in the truly white-box setting when the attacker is assumed to have full information of the methods used. Second, we propose and demonstrate the Neural Fingerprint approach for the creation of large detector banks, and application of randomized attack detection.

The rest of the paper is organized as follows: In the next section we briefly review the main approaches used for adversarial attacks, robust classification and detection of attacks. In Section (3) we present the proposed method of Neural Fingerprints for adversarial attack detection. In Section (4) we review the related work and highlight the similarity and differences from this work. Next, in Section (5) we present an evaluation of the proposed method on the ImageNet dataset, followed by a short summary and conclusion in Section (6).

## 2 BACKGROUND: ADVERSARIAL ATTACKS AND DEFENCE STRATEGIES

In this section we briefly review the most common and effective methods to create adversarial images, and the state of the art in protecting from such attacks.

### 2.1 ADVERSARIAL ATTACKS

We begin by setting the stage for both attack and defence methods. We assume that the attacked model $f(\cdot; \theta)$ is a classifier that gets an image $x$ and returns a probability vector $\hat{y}_f(x)$ over a predefined set of labels $L$. We will denote the output of the classifier by $\hat{c}_f(x)$, namely the class for which $\arg\max \hat{y}_f(x)$ is obtained. A *white-box attack* is the setting in which the network parameters $\theta$ are known to the attacker, whereas in a *black-box attack* the attacker has only oracle-access to the model.

An adversarial attack is comprised of the following steps: First, a clean image $x$ is selected with the corresponding label $c$. In the **targeted** case, the attacker has a specific desired output $c'$, whereas in the **untargeted** case the objective of the adversary is simply to change the output to be anything other than $c$. The adversary generates an altered image by introducing a minor perturbation $\eta$:

$$x' = x + \eta \tag{1}$$

Finally, the attack is deemed successful if the perturbation $\eta$ is imperceptible for the human eye, but the classification $\hat{c}_f(x')$ gives the desired output. That is, if $\hat{c}_f(x') = c'$ in the targeted case, or simply $\hat{c}_f(x') \neq c$ in the untargeted case.

The perturbation $\eta$ used to create the adversarial image $x'$ must be imperceptible to humans. This constraint is typically operationlized by limiting the magnitude of the perturbation $\eta$ under some metric, to ensure that the difference between the original input $x$ and the perturbed input $x'$ remains below a certain threshold $d$. Often, this distance $d$ is measured using an $L_p$ norm so as to emphasize pixels with large deviations.

The most direct way to obtain an adversarial example is through gradient-based methods. This broad category utilizes gradient information to determine the direction of perturbation that maximizes the desired output of the model. Let $l$ be a loss function for the model as a function of the input image. For example, the standard cross entropy loss with respect to the desired target class $c'$:

$$l(x) = -\log \hat{y}_f(x)[c'] \tag{2}$$

where $\hat{y}[c']$ is used to denote the $c'$-th element of the output $\hat{y}$. The gradient $-\nabla_x l(x)$ gives the direction of movement *in pixel space* needed to produce a targeted adversarial example. Essentially, this is the same procedure as when training the model, except that the input and parameters switch roles. During training, the training data is fixed and model parameters are updated according to the gradient of the loss function with respect to the parameters, whereas when computing the adversarial perturbation the parameters are fixed and the input image is updated according to the gradient with respect to the pixels. Most adversarial attack methods follow this logic either explicitly or via various

workarounds (which are required for instance when direct access to to the parameters, and hence the gradients, is not available).

The Fast Gradient Sign Method (Goodfellow et al., 2014) is a technique for generating adversarial examples by conducting a single gradient step:

$$\eta = -\alpha \cdot \text{sign}\left(\nabla_x \, l(x)\right) \qquad (3)$$

Where, $\alpha$ is a small constant controlling the magnitude of the perturbation, and $\text{sign}(\cdot)$ computes the sign elementwise. This technique has been found to reliably cause various neural network models to misclassify their input data.

Iterative FGSM (IFGSM) (Dong et al., 2018) applies the FGSM step multiple times within an $L_\infty$ bound $\epsilon$ on the total perturbation. That is, it repeats the following update:

$$x_{i+1} = Clip_\epsilon \left(x_i - \alpha \cdot \text{sign}(\nabla_x \, l(x))\right) \qquad (4)$$

where $\text{Clip}_\epsilon(x) = \min(\max(x, x_0 - \epsilon), x_0 + \epsilon)$. The step size $\alpha$ is in this case typically smaller than the total budget $\epsilon$ so that IFGSM can make multiple steps without exceeding the constraint. The iterative application of FGSM allows IFGSM to account for gradient directions that may not be directly toward the decision boundary from the starting point. By accumulating these gradient steps, it can find adversarial examples that FGSM would not acheive in a single step. More generally, projected Gradient Descent (PGD) attacks (Madry et al., 2017) take multiple small gradient steps while projecting back onto the allowed perturbation set after each step. The different approaches in this family differ mostly by the metric used in the projection.

Black box attacks mostly follow the same general logic, except that the direct computation of gradients is no longer possible. The Substitute Blackbox Attack (SBA) (Papernot et al., 2016) method starts by querying the model on a set of inputs and training a substitute model on this dataset. Next, a white-box attack is performed with the substitute model. Other methods use numeric estimates of gradients (Spall, 1992; Chen et al., 2017) which are plugged into the standard whitebox methods.

## 2.2 DEFENCES

We now turn to discuss methods for defence against adversarial attacks. Broadly speaking, adversarial defences are grouped into two main approaches: improving model robustness and detecting adversarial inputs. Adversarially-robust methods aim to produce the correct output whether presented with clean or attacked inputs. Detection methods are applied in parallel (or prior to) the main deep learning model, and aim to classify the input into clean versus attacked.

Adversarial training aims to improve robustness by incorporating adversarial examples into the training data. The model is trained on original examples plus versions of those examples perturbed with adversarial attacks. This exposes the model to adversarial inputs during training. To generate these adversarial examples, one can adopt various techniques such as the Fast Gradient Sign Method (FGSM) (Goodfellow et al., 2015).

One drawback of this method is that models trained in this manner remain susceptible to other forms of adversarial examples not encountered during the training process. Another approach is the use of a Barrage of Random Transforms (BaRT), as proposed in the study by Tramer et al. (Tramer et al., 2017). The key idea behind BaRT is to combine dozens of individually weak defences into a single strong defence that is robust to attacks. Specifically, BaRT applies many random image transformations like color reduction, noise injection, and FFT perturbation to each input image before classification. While each transform alone can be defeated, together they provide a boost in adversarial robustness.

A key consideration when applying adversarial training or other adversarial defences is the potential impact on accuracy of clean, unperturbed inputs. Hardening a model against adversarial attacks requires some trade-off with performance on the original task. As Kurakin et al. (Kurakin et al., 2016) found a moderate decrease in clean image accuracy when adversarial training was applied, with more robust models generally exhibiting a larger drop. In general, adversarial training induces some minor accuracy drop in order to gain improved robustness, typically around 1%, as reported by Kurakin et al. (Kurakin et al., 2016).

Adversarial detection defences aim to detect adversarial examples by training networks to distinguish between legitimate and adversarial inputs. A key advantage of adversarial classification is that it can be applied to any pretrained model without needing to modify the model architecture or training process. However, a core challenge is enabling the detector to generalize across diverse perturbation types and datasets.

A common approach for adversarial classification is to augment a classifier network $f(x)$ with a detector network $D(x)$ that outputs a prediction $y_{adv}$ or $y_{clean}$ indicating whether the input $x$ is adversarial or clean (Metzen et al., 2017). The detector network $D(x)$ is trained on a dataset containing a mix of clean examples from the original training data and adversarial examples specifically crafted to try to evade detection by the network.

Several works have proposed training feedforward neural networks as detector models. The detector network $D(h(x))$ typically takes the activations of an intermediate layer $h$ of the classifier network $f(x)$ as input rather than the raw input data (Metzen et al., 2017).

## 3 THE METHOD OF NEURAL FINGERPRINTS

In the pure white-box setting even a near-perfect deterministic adversarial attack detector is insufficient. By pure white-box we mean that the attacker has full knowledge and access to the model and trained parameters, as well as the detector model that will be employed to detect the attack. By deterministic we mean that the defence model is constant, so the attacker is able to check offline weather or not each adversarial input created will be detected. Consider for instance a near-perfect deterministic adversarial attack detector with a detection rate of 99.9%. The attacker generates a few thousand unrelated adversarial inputs, finds on average a few that pass the defence, and discards all others. Then, the online system is only fed these few inputs that are already known to fool the defence.

What is needed for an effective defence in this case is a large family of detectors from which one (or more) is sampled at random in real time for each input. The size of the family should be large enough so that an attacker is not able to find inputs that fool them all (even if such inputs exist). This randomization assures us that the attacker is not able to work offline to find inputs that are known to fool the detector. For the defender on the other hand the computation needed to generate a sufficient number of detectors must be feasible. Finally, the detectors should have adequately good detection properties.

### 3.1 NEURAL FINGERPRINTS

Consider a deep neural network classifier $f(x; \theta)$ with parameters $\theta$. For an input image $x$, let $A(x) \in \mathbb{R}^N$ denote the concatenated vector of activations for the last $\ell$ layers, containing in total $N$ neurons. That is, the last $\ell$ layers are of sizes $n^1, n^2, \ldots, n^\ell$, and $\sum_i n^i = N$. A $d$-size *fingerprint* is a subset $S \subseteq \{1, ..., N\}$ of size $d$ that indexes into $A(x)$. We use $A(x, j)$ to denote the activation of the $j^{th}$ neuron in $A(x)$. The *fingerprint value* is given by:

$$F_S(x) = \frac{1}{d} \sum_{j \in S} A(x, j) \tag{5}$$

We generate $K$ fingerprints $S_1, S_2, \ldots, S_K$ where $K$ is a hyper-parameter of the method. The procedure used to generate the fingerprints is described at the end of this section. In total, this defines a $K$-dimensional feature representation of the input $x$:

$$\Phi(x) = [F_{S_1}(x), \ldots, F_{S_K}(x)] \tag{6}$$

The main goal is to model $P_{\text{clean}}(\Phi(x)|y)$, the distribution of fingerprints conditioned on each of the classes $y$, and $P_{\text{attack}}(\Phi(x)|y)$, the distribution of fingerprints conditioned on inputs from another class being adversarially attacked to class $y$. At test time, for an input $x$ and the associated prediction $\hat{c}_f(x)$, all that remains is to determine if the feature vector $\Phi(x)$ is likely under the predicted class $\hat{c}_f(x)$. To this end, we define the likelihood functions for the observed input under clean and attacked class models:

$$\mathcal{L}_{\text{clean}}(y \mid x) = P_{\text{clean}}(\Phi(x) \mid y) = \prod_{i=1}^{K} P_{\text{clean}}(F_{S_i}(x) \mid y) \tag{7}$$

$$\mathcal{L}_{\text{attack}}(y \mid x) = P_{\text{attack}}(\Phi(x) \mid y) = \prod_{i=1}^{K} P_{\text{attack}}(F_{S_i}(x) \mid y) \tag{8}$$

where the second equality in each case stems from an independence assumption for the fingerprints, and $P_{\text{clean}}(F_{S_i}(x) \mid y)$ and $P_{\text{attack}}(F_{S_i}(x) \mid y)$ are density models for the $i$-th fingerprint with clean and attacked predicted class $y$ respectively. The decision rule is then a threshold on the likelihood ratio:

$$\frac{\mathcal{L}_{\text{clean}}(y \mid x)}{\mathcal{L}_{\text{attack}}(y \mid x)} > \alpha \tag{9}$$

or equivalently using the log likelihoods:

$$\sum_{i=1}^{K} \log \left( P_{\text{clean}}(F_{S_i}(x) \mid y) \right) - \sum_{i=1}^{K} \log \left( P_{\text{attack}}(F_{S_i}(x) \mid y) \right) > log(\alpha) \tag{10}$$

The set of decision rules for possible values of $\alpha$ defines an ROC curve, and a point can then be selected that achieves the best possible detection while maintaining an acceptable rate of false positives. In addition to the likelihood ratio threshold test, we propose two additional decision rules. The first is a simpler version of aggregating individual fingerprint information via a vote between them. That is, using a threshold on on the number of votes for flagging an attack:

$$\sum_{i=1}^{K} \mathbb{1} \left[ P_{\text{attack}}(F_{S_i}(x) \mid y) \geq P_{\text{clean}}(F_{S_i}(x) \mid y) \right] \tag{11}$$

One drawback of both the likelihood ratio and voting tests is that they require the distribution of fingerprint values for attacked images as well as the claen ones. When this is not available, it is

possible to resort to an anomaly detection approach, setting a threshold on the likelihood under the clean model only, that is:

$$\sum_{i=1}^{K} \log \left( P_{\text{clean}}(F_{S_i}(x) \mid y) \right) \tag{12}$$

Two final remaining element are the estimation of the density functions $P_{\text{clean}}(F_{S_i}(x)|y)$, $P_{\text{attack}}(F_{S_i}(x)|y)$, and the method used to obtain effective fingerprints. Recall each fingerprint is the average of many neuron activations from different parts of the network, and it stands to reason that these will be almost completely independent, conditioned on the predicted class. Hence, it is sufficient to approximate the individual fingerprint density functions using a Gaussian approximation. This was verified empirically (See figure 1).

For efficient computation of fingerprints we suggest the following preprocessing. For a specific class $c$, we begin with a set of $m$ images from the class, and $m'$ random images of other classes, which underwent a targeted adversarial attack and are now classified as class $c$. All images are then fed through the model, and the $N$ activations that are considered for fingerprint membership are stored for each one. In total this produces a table of size $N \times m$ for the clean images, and $N \times m'$ for the attacked images. The remaining computation requires only these tables (the images and model are no longer used).

The procedure we use to generate effective fingerprints is based on sampling and filtering. At each step a fingerprint (that is $k$ out of the total $N$ activations in the tables) is sampled. Next, the parameters (mean and variance) for the Gaussian approximation of the distribution of the fingerprint value is computed for clean and attacked images. Finally, a Cohen's d (Cohen, 2013) effect size is calculated to determine the usefulness of the fingerprint in separating the clean and attacked inputs. If the fingerprint's effect size is above a pre-determined threshold it is added to the fingerprint bank. We note that in the anomaly detection variant the filtering step is not possible, as we are only using the clean images, and hence all fingerprints are used.

## 4 RELATED WORK

Neural probing is a method originally developed to ascertain how suitable the representation in each layer of a deep neural network is for the purpose of the learned task (Alain & Bengio, 2016), and has since become a fundamental tool for understanding models in natural language processing (Belinkov, 2022). In this method, entire representations (hidden layers) are used as features for a predictor of some aspect of the input, and the success of the predictor is understood to measure how well the aspect is encoded in the representation. Using this framework, each fingerprint in our work can be described as a random sparse linear readout for the binary prediction of adversarial attacks. The success of these predictors points to the idea that the presence of adversarial attacks is encoded in the representation of the final layers of the model. However, clearly using a straightforward probe from these layers is insufficient for protection, as an attacker could simply add this additional output to the objective of the adversarial attack, or find inputs that are not detected via offline trial and error. It is the combinatorical size of the fingerprint space as they are formulated here, that provides some additional protection from straightforward workarounds (see Section 3).

Several methods have previously utilized hidden layer activations for detection of adversarial attacks or for robust classification. For example, in (Zheng & Hong, 2018) entire hidden layers are modeled using Gaussian Mixture Models (GMMs), and an adversarial attack is declared if the likelihood of an image in the fitted GMM is below a threshold. In (Feinman et al., 2017) a similar method is used based on a Kernel Density Estimate (KDE) model of the last hidden layer.

**(a) Randomly selected fingerprints**       **(b) Filtered fingerprints**

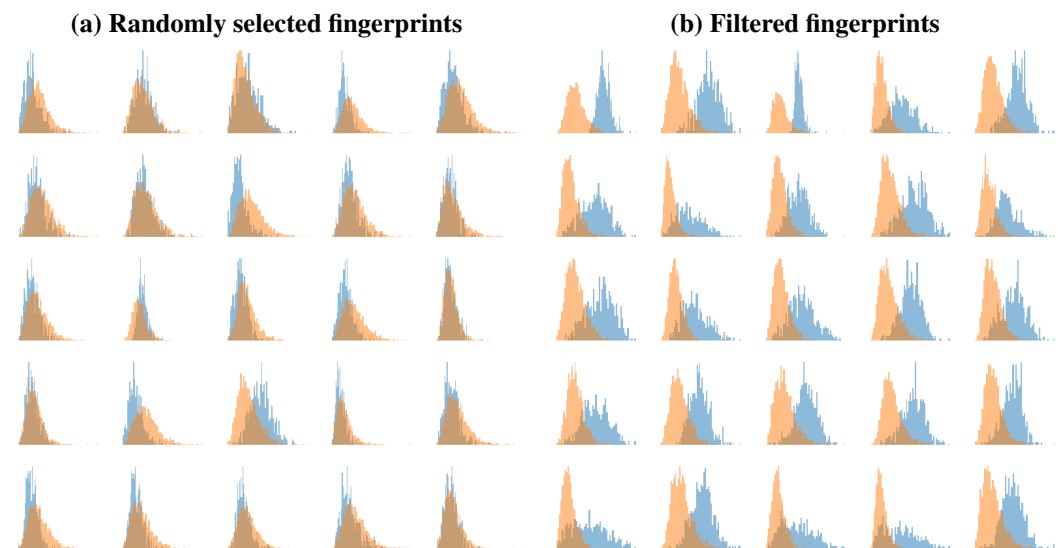

Figure 1: Fingerprint distributions: (a) randomly selected fingerprints (b) fingerprints filtered based on effect size. Orange - clean image, blue - attacked.

The method proposed here differs from all the above in two fundamental aspects. First, and most importantly, all the above inherently employ static functions of the network activations that can be added to the attack objective, while the possibility of randomization with the method proposed here offers some protection from this straightforward bypass. Second, our method, using only sparse linear combinations of activations, is fast and easy to implement with any existing network structure.

## 5 RESULTS

In this section we evaluate the proposed method of Neural Fingerprints, and compare the three decision rule alternatives of likelihood ratio, voting, and anomaly detection (see Section 3). To the best of our knowledge this is the first randomized method for detection of adversarial attacks operating under the assumption that the attacker has full knowledge of the system, hence the purpose of this evaluation is only to determine the feasibility of the method, as there are no relevant alternatives to compare to. For this purpose we use the ImageNet validation data.

We repeat the following for each of the tested deep learning models and attack methods: from each category, 500 images were sampled, with 400 allocated for training and 100 reserved for testing. Only images with a minimum of 50% confidence for the correct class were considered for sampling. Likewise, 500 images were randomly selected from all other categories and attacked so that the model classified them as the current category (examples of original and attacked images are presented in Appendix A). The attacked images were also subdivided into 400 for training and 100 for testing. In total, this amounts to 1000 images per category and one million in total.

The network architectures used for the evaluation are Inception V3[1] (Szegedy et al., 2016) and ViT[2] (Dosovitskiy et al., 2021). These two were selected as representatives of the convolution based and transformer based model families. To obtain adversarial attacked images we used the CleverHans

---

[1] https://huggingface.co/docs/timm/en/models/inception-v3
[2] https://huggingface.co/timm/vit_base_patch16_224.augreg2_in21k_ft_in1k

| Model | Attack | Detection Method | 1% FP | 2% FP | 5% FP |
|---|---|---|---|---|---|
| Inception V3 | IFGSM | Vote | 98.2% | 99.3% | 99.7% |
| | | Anomaly | 94.0% | 97.6% | 99.3% |
| | | Likelihood Ratio | 98.7% | 99.4% | 99.7% |
| | PGD | Vote | 97.9% | 99.2% | 99.5% |
| | | Anomaly | 97.4% | 98.8% | 99.4% |
| | | Likelihood Ratio | 98.6% | 99.1% | 99.5% |
| ViT | IFGSN | Vote | 97.4% | 98.8% | 99.8% |
| | | Anomaly | 93.6% | 97.8% | 99.5% |
| | | Likelihood Ratio | 96.9% | 99.0% | 99.9% |
| | PGD | Vote | 96.8% | 98.5% | 99.3% |
| | | Anomaly | 95.0% | 97.2% | 98.4% |
| | | Likelihood Ratio | 96.5% | 97.6% | 99.4% |

Table 1: Adversarial attack detection rate for the ImageNet dataset using ViT and Inception V3 models, tested against IFGSM and PGD attacks across 3 detection methods: Vote, Anomaly, and Likelihood Ratio with 20 fingerprints.

implementation (Papernot et al., 2018) [3] of Iterative Fast Gradient Sign Method (IFGSM) and Projected Gradient Descent (PGD). For IFGSM, the attack parameters include the number of iterations (*iter=150*) and the magnitude of the perturbation (*eps=0.01*), however we terminated each attack upon reaching confidence of at least $70\%$ in the target class, which was normally achieved after 3 to 10 iterations. Similarly, for PGD, the key parameters used are (*eps=0.01*), the number of iterations (*iter=40*), and the step size (*step size=0.01*). Here also we stop whenever reaching at least $70\%$ confidence in the target class. For each iteration, $100,000$ fingerprints of size $d = 50$ were sampled, and the top 20 were selected based on the training data.

We first consider the individual fingerprints. Figure (1) shows an illustrative example of fingerprints generated for class *toucan*[4] in the Inception V3 model and IFGSM attack setting. The general fingerprints sampled (left panel) mostly show high overlap in distribution between the clean and attacked images, with a few exceptions. When sampling based on effect size (Cohen's d $> 1$) 3 (right panel) we are able to obtain fingerprints with high individual separating power. With these in mind, it is easy to see how combining many random fingerprints of this sort (either by voting or likelihood model) will result in good detection performance.

The main results are presented as test data detection rate when setting the false detection rate to $1, 3$ or $5\%$ (Table 1). First, detection rates are relatively high for all combinations of deep learning model, attack method, and detection rule, ranging from $93.6\%$ to $99.9\%$. As expected, the likelihood ratio detection rule offers the best overall performance from among the three tested approaches, followed by the voting decision rule, with the anomaly detection approach trailing behind. When considering the effect of number of fingerprints used for each input (Figure 2), we see a saturation of the detection AUC at $20 - 40$ fingerprints. Furthermore, the detection performance for the ViT model saturates higher but later than for Inception V3, suggesting that slightly increasing the number of fingerprints used in the main results Table (1) beyond 20 could be beneficial for ViT adversarial attack detection.

## 6 CONCLUSION

Deep learning models have been shown to suffer from vulnerability to adversarial attacks, which are small perturbations to the input, imperceptible to humans, but causing the model to misclassify

---

[3] https://github.com/cleverhans-lab/cleverhans
[4] n01843383

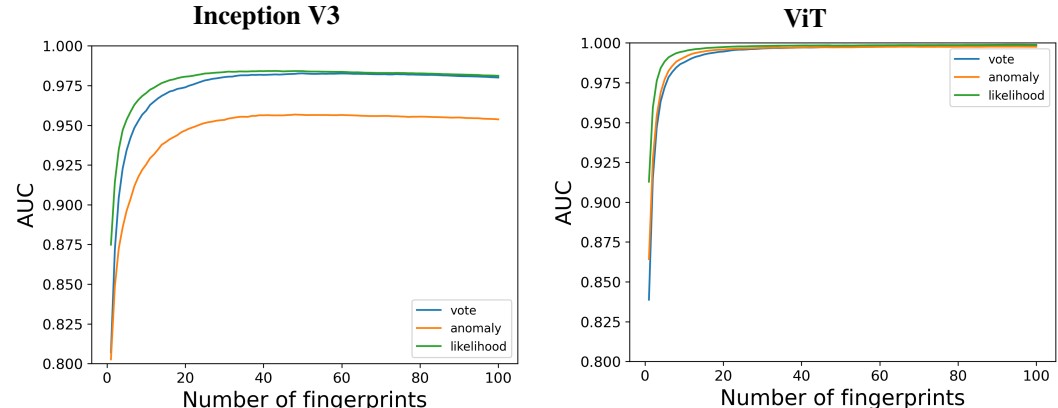

Figure 2: Detection AUC as a function of Number of Fingerprints for the three detection methods: Vote, Anomaly, and Likelihood for both models.

the input. Although existing adversarial attack detection methods often have excellent performance, we argue this is not enough. In the truly white-box setting, when the attacker knows the structure and parameters of the classifier and detection models, a deterministic system with less than perfect accuracy will not suffice. To overcome this inherent limitation, we suggest the method of Neural Fingerprints for creating a large bank of attack detectors, from which a few can be sampled for each input at test time. The simplicity and scalability of this approach enables us to build a very large bank of detectors to sample from, so that the straightforward attacks against any deterministic system (see Section 1) are no longer feasible. Results conducted on the ImageNet dataset with standard deep learning models and adversarial attacks shows the efficacy of the proposed method with high detection rates and a low proportion of false positives.

In this work we suggest to combine the individual fingerprints that are sampled for each input using a likelihood ratio test. Treating them as week classifiers, future work will address the question of improving on the results presented here via a boosting framework. Another possible extension is the use of the same idea for robust classification rather than detection. Finally, the Neural Fingerprint method is presented and tested in this paper in the language of image classification, but is not inherently limited to this domain.

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

# 7 APPENDIX A: ATTACKED IMAGES

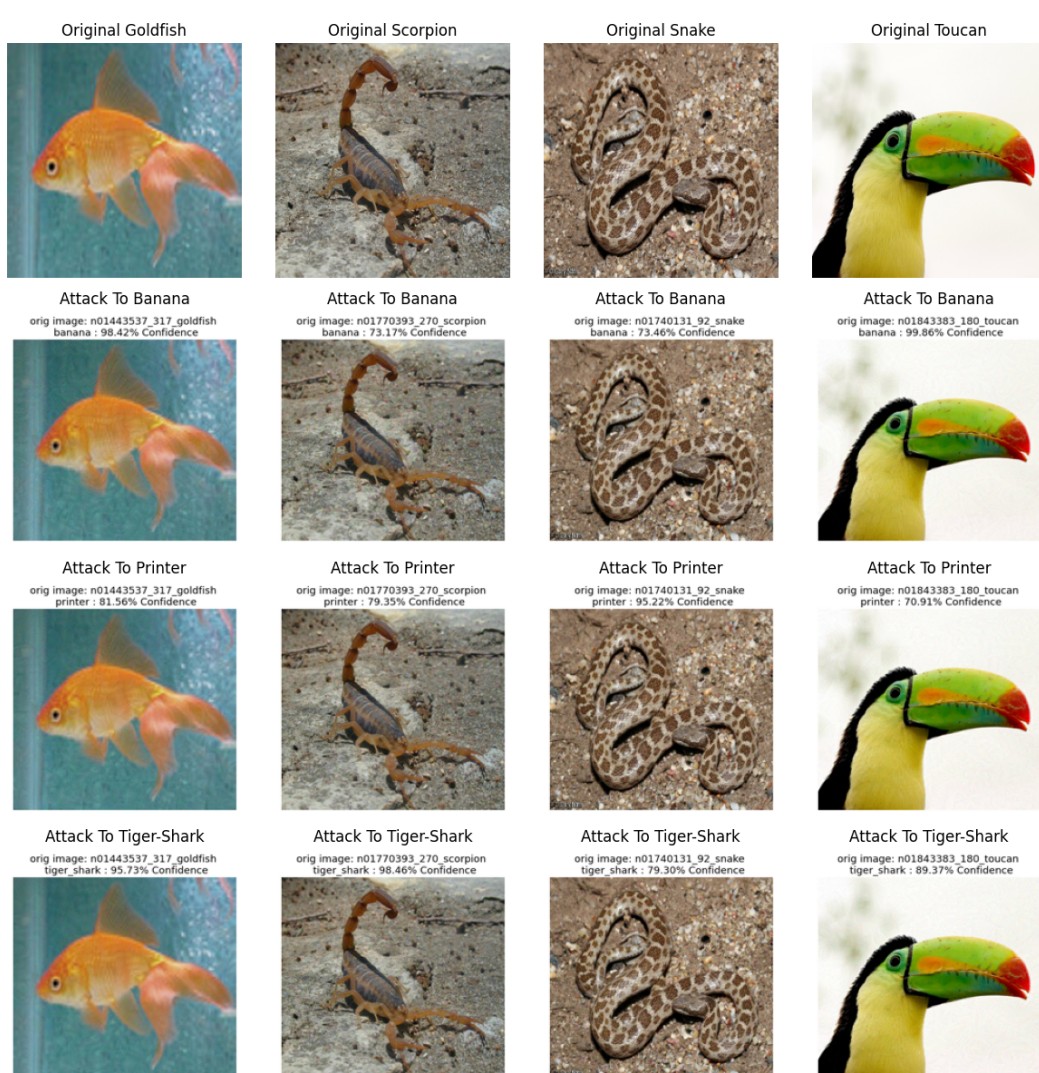

Figure 3: Example of original images and their adversarial attacks. The first row shows original images of classes goldfish, scorpion, snake, and toucan. The subsequent rows demonstrate IFGSM attacks on Iception V3, of each original image to three other categories: Banana, Printer, Tiger-Shark.

