# OpenReview forum: "Neural Fingerprints for Adversarial Attack Detection"
_ICLR.cc/2025/Conference — ICLR 2025 Conference Withdrawn Submission_

### Official Review · Reviewer_xJ2F · 2024-10-26

**Soundness:** 1
**Presentation:** 3
**Contribution:** 1
**Rating:** 3
**Confidence:** 4

**Summary:**

Adversarial machine learning is a pressing concern for the deployment of AI in safety critical settings. Adversarial examples (images with small malicious input manipulations) remain a cogent threat. This paper proposes a way to mitigate adversarial examples through a novel detection approach. Specifically, from the classifiers a random set of neurons is selected and used to make a “fingerprint”. During test time an input sample is compared with a bank of fingerprints in a likelihood ratio test to detect if the sample is adversarial or not. Empirically, the methods effectiveness is demonstrated on ImageNet.

**Strengths:**

The idea is novel and I like that the defense includes some element of randomization. In general randomization as the underlying property of a defense seems to be a slightly more solid basis for security (although I will mention why that is not always the case in the weakness section). Having tested on ImageNet is a very important metric for any new adversarial defense.

**Weaknesses:**

I will preface my critique of the paper by saying that I don’t think this is an inherently bad work, however there are a number of critical issues that need to be done to strength the paper.

Issue 1: The threat model/white-box attack analysis is utterly lacking. The authors claim to test on IFGSM/PGD but this is a VERY old white-box attack. Why do the authors not test on more state-of-the-art white-box attacks like APGD? https://arxiv.org/pdf/2003.01690

Issue 2: There are no black-box attacks. The authors also mention Papernot’s 2016 transferability attack but I don’t see any results where they test on it. In addition, Papernot’s attack has been enhanced in this work: https://arxiv.org/abs/2006.10876 The authors should test the enhanced transfer based black-box attack.

Also they should not just consider black-box transfer attacks, but query based attacks like RayS from this paper: https://arxiv.org/abs/2006.12792. Now one might question, if the threat model is white-box then why do the authors need to consider black-box attacks? Well often times gradient masking or obfuscated gradients lead to false reports of robustness (https://arxiv.org/abs/1802.00420). Therefore, even if the threat model is white-box, the security analysis is not complete without black-box empirical results.

Issue 3: No code is given by the authors that I can find. How can we trust that other authors will be able to recreate their results without any code in the form of an anonymized Github or zip file?

Issue 4: No comparison to other detection methods. There are at least TWO other detection papers that exist that the authors should cite and compare their work to:

https://arxiv.org/abs/1902.04818

https://ieeexplore.ieee.org/document/9663375

As I am not an expert in adversarial detection, I am sure there may be additional works, but the bare minimum would be to cite the above two papers and either experiment with them, or put in the related work why or why not these methods merit comparison.

Issue 5: Single dataset experiments. The authors ONLY test on ImageNet. About five years ago when the Barrage of Random transforms paper was published, they also only tested on ImageNet. However, for modern papers, a single dataset is not sufficient. As the reviewer I need to be convinced that the method is applicable and generalizable. With single dataset experiments, I am simply not convinced.

Issue 6: Adaptive attacks. In https://arxiv.org/abs/2002.08347 they show how many white-box defenses can be broken by adapting the attack to the defense. The authors do not have an discussion on adaptive attacks or how their method might be able to be overcome with a more specific and customized white-box attack on their defense.

**Questions:**

I am sorry to have to be so negative in my weakness section of the review. I think the authors have an interesting idea but I would like to see it fully fleshed out with the proper attack/defense evaluations and more than one dataset. My question is can you address issues 1-6 that I mentioned in my weakness section in a satisfactory manner? Thank you.

---

### Official Review · Reviewer_9yKx · 2024-10-28

**Soundness:** 1
**Presentation:** 2
**Contribution:** 2
**Rating:** 3
**Confidence:** 4

**Summary:**

The authors propose a novel method called "Neural Fingerprints" to improve adversarial attack detection by randomly sampling from a large bank of detectors at test time. While the idea shows promise, the reviewer believes that the paper is not yet mature and requires further development. Specifically, the reviewer suggests that the authors should:

 - Strengthen the related work section to better motivate their method
 - Conduct more experiments to support their claims
 - Discuss the limitations of their approach and potential avenues for future work

The reviewer notes that the paper is well-written and concise, but feels that it falls short of its full potential. They look forward to seeing the authors' revisions and the opinions of other reviewers.

Recommendations for Authors:

 - Clarify the motivation for assuming a complete white-box setting
 - Provide more context on the history of adversarial attacks and discuss other attacks, such as [1]
 - Clearly define "deterministic attacks" and discuss the strengths and limitations of the proposed method
 - Improve the experiment quality by reporting more metrics and using standard values for epsilon size

[1] https://attackbench.github.io

**Strengths:**

Their method shows strong performance on ImageNet and larger DNNs architectures with high detection rates and low false positives using standard deep learning models. The paper suggests future improvements through likelihood ratio tests and boosting frameworks, while noting the approach could extend beyond image classification to other domains.

**Weaknesses:**

- 022: The motivation is unclear to me: Why is it common security setup o to have complete knowledge?  think that companies try to protect models against model stealing and make it difficult to have full access of the model.
It is ok to have complete white-box setting but the statement that is common in security does not seem adequate to me.

- Introduction/Related work: The threat “adversarial examples” is not a new phenomenon. The first attack (FGSM) was published in 2014/2015 by Goodfellow. It has been around 10 years now that these models existed.
I think that the authors should have been discussing in the related work why only focusing on FGSM and its variant.  PGD  attack is just shortly mentioned.
There are no words for other adversarial attacks [1].

- The authors claim that it works on deterministic attacks.  The PGD attack is by my knowledge deterministic.
The definition of “determinisitic attacks” could be better worked out for better understanding of the strengths and limitations of the proposed method.

- Experiment quality: Only the  False Positive Rate  (FP) is stated in Table 1. The attack epsilon size is selected 0.01. For ImageNet the standard epsilon size for this attack  methods would be 4/255 .
A ResNet18 or WideResNet28-10 are very often used neural networks, which are not used in this setup. Are these one too small? Could this be a limitation?

**Questions:**

Transferability to other models or datasets: The strength of adversarial examples (even from black-box attacks) is that they transfer well to other models. Can your method generate the fingerprints on model A but defend model B?

---

### Official Review · Reviewer_dwyc · 2024-11-02

**Soundness:** 2
**Presentation:** 1
**Contribution:** 2
**Rating:** 3
**Confidence:** 4

**Summary:**

This paper proposes an adversarial example detection method based on analyzing fingerprints that are average values of $d$-size random subsets of activations of neurons from the last $l$ layers. The intuition behind this is inspired by the previous work, neural probing, saying the presence of adversarial attacks is encoded in the representation of the final layers of the model. Detecting an adversarial example is based on three rules: likelihood ratio, voting, and anomaly detection. The likelihood ratio and voting rules require attacked images in the analysis to decide whether or not an input image is an adversarial example. The proposed detection method is validated on ImageNet validation data (500 clean images, 500 attacked images) with Inception V3 and ViT. The paper uses IFGSM and PGD for attacks and shows high detection results on 3 decision rules: likelihood ratio, voting, and anomaly detection.

**Strengths:**

- The paper points out that existing adversarial detection methods are not fully white-box. The paper reasons that a single detector scenario is not sufficient. If random unknown detectors are used, it will be hard for an attacker to generate a successful adversarial example to bypass the detection method.
- The paper brings an interesting insight into fingerprint distribution in Fig. 1, where fingerprints of clean images and attacked ones are less overlapped, showing the separation ability.

**Weaknesses:**

- The paper explains the proposed ideas with the analogy of detectors, and detectors and fingerprints seem interchangeably used. This causes a bit of confusion. To improve clarity, please provide explicit definitions of "detectors" and "fingerprints" early in the paper, and to consistently use these terms throughout. This would help readers better understand the key concepts.
- In Section 2.2, BaRT is not proposed by Tramer et al. (It is Raff et al. CVPR 2019).
- The paper stated that the proposed method is the first randomized detection method under full white-box settings, and there is no comparison with state-of-the-art detection methods. Therefore, it is difficult to evaluate the proposed method.
Please compare the proposed method to existing state-of-the-art detection methods, even if those are not randomized. This would provide context for the performance of the proposed approach. In addition, please discuss potential ways an attacker might try to circumvent the proposed randomized approach, to better evaluate its robustness.
- Experiments are limited. The paper only deploys IFGSM and PGD, which are similar attack methods.
Please consider additional attack methods such as CW, EAD, SPSA, N-Attack, AutoAttack (suite of attacks). This would help assess the generalizability of the proposed approach across a wider range of adversarial attacks.

**Questions:**

1. In general, detection methods do not know how adversarial examples are generated. In this scenario, the proposed method uses anomaly detection. How to find the threshold for anomaly detection? How do we know the threshold is generalizable for different attacks? It is better to provide the explicitly defined threat model for the proposed method so that we know to what extent it can detect adversarial examples with what guarantee (reliability).
2. In Section 3.1, Eq. 7 and 8, the input $x$ is always the same for both clean and attacked distribution. Shouldn't they be different? The other notations also use the same $x$ for everything.
3. In Section 3.1, $N$ is used for the total number of neurons. However, at the end of Section 3.1, $N$ is again used for a number of qualified fingerprints to be stored. It would be better to have clear notations.
4. How the results in Table 1 are obtained? Do you use only one detector bank containing fingerprints of clean images and attacked images (IFGSM and PGD) for three decision rules? Or you need a separate detector bank for each attack method. Are the results obtained independently?
5. How do we know if the proposed method can be generalized to unseen attacks? Please consider a wide range of attacks such as CW, EAD, SPSA, N-Attack, AutoAttack to perform a cross-validation study across different attack types.
6. How will Figure 1 look like if you use attacked images that CW or EAD generates? What about black-box attacks?
7. Is there any relationship between noise budget and detection performance? In experiments, only one fixed budget of 0.01 was used.
Please conduct experiments with varying noise budgets and report how detection performance changes. This would provide insight into the robustness of the method across different attack strengths.

### Minor Comments
- In Section 5, last paragraph, the false detection rate, 1, 3, and 5 % are used. But Table 1 shows 1, 2, and 5%. Maybe it is a typo.
- Current explanations are not very reproducible. Please provide implementation details and threshold parameters.

---

### Note · Authors · 2024-11-25

**Comment:**

Hi,

We hope this message finds you well. We would like to request the withdrawal of our submission.

After further consideration, we have determined that additional time and computational resources are needed to extend our results to include evaluations on more attacks and datasets. This improvement will enhance the robustness and comprehensiveness of our study, but unfortunately, we cannot complete this work within the current conference timeline.

We deeply appreciate the opportunity to submit to ICLR and apologize for any inconvenience caused by this decision. We are grateful for your understanding and look forward to potentially resubmitting a more complete version of our work to future venues.

Thank you for your time and consideration.

**Withdrawal Confirmation:**

I have read and agree with the venue's withdrawal policy on behalf of myself and my co-authors.